# Home Parenteral Nutrition for Children: What Are the Factors Indicating Dependence and Mortality?

**DOI:** 10.3390/nu15030706

**Published:** 2023-01-30

**Authors:** Ying-Cing Chen, Chia-Man Chou, Sheng-Yang Huang, Hou-Chuan Chen

**Affiliations:** 1Department of Surgery, Taichung Veterans General Hospital, Taichung 407219, Taiwan; 2Division of Pediatric Surgery, Department of Surgery, Taichung Veterans General Hospital, Taichung 407219, Taiwan; 3School of Medicine, College of Medicine, National Yang Ming Chiao Tung University, Taipei 112304, Taiwan; 4Division of Post-Baccalaureate Medicine, College of Medicine, National Chung Hsing University, Taichung 402202, Taiwan

**Keywords:** parenteral nutrition, parenteral nutrition-associated liver disease (PNALD), parenteral nutrition-associated cholestasis (PNAC), intestinal failure-associated liver disease (IFALD), central line-associated bloodstream infection (CLABI), catheter-related bloodstream infection, Pediatric End-Stage Liver Disease (PELD) score

## Abstract

Parenteral nutrition (PN) in children with short bowel syndrome is crucial and lifesaving. Taking care of such patients requires interprofessional practice and multiple team resource management. Home PN (HPN) usage allows patients and families to live regular lives outside hospitals. We share our experiences for the last two decades and identify the risk factors for complications and mortality. A retrospective study of HPN patients was conducted between January 2000 and February 2022. Medical records of age, body weight, diagnosis, length of residual intestines, HPN period, central line attempts, complications, weaning, and survival were collected and analyzed. The patients were classified as HPN free, HPN dependent, and mortality groups. A total of 25 patients received HPN at our outpatient clinic, and one was excluded for the adult age of disease onset. There were 13 patients (54.1%) who were successfully weaned from HPN until the record-enroled date. The overall mortality rate was 20.8% (five patients). All mortality cases had prolonged cholestasis, Child Class B or C, and a positive Pediatric End-Stage Liver Disease (PELD) score. For HPN dependence, extended resection and multiple central line placement were two significant independent factors. Cholestasis, Child Class B or C, and positive PELD score were the most important risk factors for mortality. The central line-related complication rate was not different in all patient groups. The overall central line infection rate was 1.58 per 1000 catheter days. Caution should be addressed to prevent cholestasis and intestinal failure-associated liver disease during the HPN period, to prevent mortality. By understanding the risks of HPN dependence and mortality, preventive procedures could be addressed earlier.

## 1. Introduction

Intestinal failure, under which nutritional requirements cannot be met, results in the need for parenteral nutrition (PN) support. Currently, individually tailored PN solutions are generally used as the primary treatment for pediatric patients, especially neonates and infants. The prevalence of home PN (HPN) in children has doubled from 2012 to 2019 in the United Kingdom [1]. Various indications of home total PN (TPN) for children are mentioned, including colitis, inflammatory bowel disease, intestinal obstruction, motility disorder, malnutrition, and malignancy [2]. These observations differ from those of adult patient groups with HPN support. Based on the unpublished data of the Nutrition Therapy Team of Taichung Veterans General Hospital, more than 90% of the 44 new cases in 2021 are intrabdominal malignancy. Historically, only 26.7% of adult patients resume enteral nutrition, and the mortality rate is as high as 65%. Conversely, the common indications for pediatric HPN in our institute were short bowel syndrome (SBS), intestinal failure (IF), and motility disorder. The etiology of SBS is either congenital or acquired.

Given the increasing requirement of long-term HPN use, central venous access with the relevant infectious, thrombotic, and mechanical complications has been emphasized in clinical care and literature. For example, the incidence of catheter-related bloodstream infection (CRBSI) in Croatia was 1.15 per 1000 catheter-days from 2011 to 2019. Among them, nosocomial infection accounted for 2.35 per 1000 catheter-days, and HPN-related infection was 0.48 per 1000 catheter-days [3]. By implementing adequate central venous catheter (CVC) care protocols, CRBSI could be reduced and managed to save lives [4]. By preventing catheter-related infection, the prolonged and safe usage of HPN for pediatric patients is possible. Some patients even keep HPN usage until adulthood.

Stanko and colleagues demonstrated the association between liver injury and the extent of gut resection, indicating that the extent of intestine loss is an independent factor in inflicting liver damage [5]. In patients with PN support, the common IF-associated liver disease (IFALD) may be monitored with difficulty using regular blood biochemical laboratory studies or hepatobiliary ultrasounds. However, improvements in IFALD management enable fewer patients to develop end-stage liver disease over the past 15 years [6]. Regarding the pathogenesis of IFALD, the mechanism and interaction between the underlying condition such as extensive small bowel resection, catheter-related infection, and PN mixture component are still unclear [7,8]. In addition to PN-associated liver disease, PN-associated cholestasis in children has been addressed within the guidelines issued since 1997 [7,8,9,10].

In the present study, the experience of HPN at our institute in the past two decades was summarized. The main risk factors of PN-related complications, dependence on HPN, and mortality were identified.

## 2. Materials and Methods

Between January 2000 and February 2022, 25 patients who received HPN support in our department were retrospectively collected. One patient was excluded because IF occurred during adulthood. All enroled patients had HPN support and regular follow-up at our outpatient clinic until weaning PN, death, or the last known clinical encounter near the end of study period. Patients survived without HPN were defined as HPN-free group. Patients dependent on HPN until the latest follow-up were defined as HPN-dependent group. Patients who did not survive during the study period were defined as mortality group. No cases of mortality group experienced HPN-free survival period after TPN initialization. All subjects enroled in this study were approved by the Institutional Committee on Human Research of TCVGH (Institutional Review Board TCVGH No. CE22266B).

Medical records were reviewed, and data were collected. The demographic factors included age of onset, sex, birth body weight (BW), BW change post-HPN, underlying disease, and the diagnosis of SBS or IF. Blood biochemistry data included bilirubin levels, alanine aminotransferase (ALT), aspartate aminotransferase (AST), albumin, and international normalized ratio. Surgical intervention procedures and the anatomy of the gastrointestinal tract were documented. For IFALD evaluation, the Child–Turcotte–Pugh classification, and Pediatric End-Stage Liver Disease (PELD) scores were documented for patients younger than 12 years of age. For patients beyond 12 years of age during the end of the study period, a Model for End-Stage Liver Disease (MELD) score was used instead of PELD. The clinical outcomes were HPN period, times of hospital stay, duration of hospital stay, complications, and survival. Elevated levels of serum AST and ALT were defined as any time that the individuals had AST or ALT data two times more than the normal upper limit. Cholestasis was defined as individuals with direct bilirubin levels that were greater than 2.5 mg/dL. Low birth weight was defined as birth BW below 2500 g. CRBSI was defined as fever with more than two positive blood-culture samples without other possible infectious sources according to The Centers for Disease Control and Prevention (CDC). Follow-up periods were calculated from the date of initial first diagnosis to death or the latest follow-up date at our outpatient clinic.

Data analysis was performed using MedCalc^®^ Statistical Software version 20.2 (MedCalc Software Ltd., Ostend, Belgium; https://www.medcalc.org (accessed on 18 May 2022). Continuous variables were expressed as mean values (ranges), and categorical variables were expressed as numbers (percentage). Comparisons between groups were performed using one-way ANOVA for continuous variables, and the chi-squared test for categorical variables. Logistic-regression analyses were performed to determine the risk factors for HPN dependence and mortality. Survival analysis was stratified using the Kaplan–Meier method. Statistical significance was set as *p*-value < 0.05.

Based on European Society for Parenteral and Enteral Nutrition (ESPEN) suggestions, the team of HPN comprises pediatric surgeons, nurse practitioners, pharmacists, and social workers to provide home care service, education, and social and psychological assistance [11]. In Taiwan, no general practitioners are involved for these patients, owing to a high density of medical centers in a small territorial area. HPN was initiated since the day of discharge from the hospital stay that SBS or IF were first diagnosed. During hospitalization, the main care givers at home were educated, trained, and validated for HPN formula preparation, administration, and central line care. A complete PN admixture with all macronutrients and micronutrients was delivered once a week during regular outpatient visits. The formulation of PN was individualized and prepared by the pharmacists in the clean room at our institute. The formula was prescribed according to the Guidelines on Paediatric Parenteral Nutrition of the European Society of Paediatric Gastroenterology, Hepatology, and Nutrition, and the European Society for Parenteral and Enteral Nutrition [12]. Glucose was responsible for 60%–70% of energy provided by PN. The amino acid dose was 1–2 gm/kg/day. To reduce IFALD, the lipid dose was limited to 1–2 gm/kg/day if tolerated. The lipid emulsion contained soybean oil, medium-chain triglycerides (MCT), olive oil, and fish oil. The trace element additives were mostly Peditrace ^®^ with the amount of 0.5–1 mL/kg/day. Considering intestinal lengthening, only one patient underwent an intestinal transplant. No patient had serial transverse enteroplasty (STEP) or Bianchi procedure. Under periodic surveillance, regimen adjustment was based on enteral nutrition, body figure, laboratory data, clinicians’ experience, and suggestions from previous publications [13,14]. Weaning PN followed a stepwise strategy and was defined as the discontinuation of HPN without reuse until the date of the latest follow-up.

Vascular access was mostly established with a centrally placed cuffed catheter through a subcutaneous tunnel (Hickman™/Broviac™, Bard, Murray Hill, NJ, USA) according to ESPEN suggestion [11]. One patient used a peripherally inserted central catheter as the PN route. Some patients had the temporary usage of implantable vascular-access systems, and no arteriovenous fistula was created in the current series. Central line revision was performed for CRBSI, dislodgement, rupture, or occlusion.

## 3. Results

All patients that were enroled in the study had detailed data without any that were missing. The collection and analysis of the data followed the instruction of the Institutional Review Board (IRB) and the Declaration of Helsinki. The results are divided into parts as below.

### 3.1. Demographic Data

A total of 24 patients received HPN from the pediatric phase at our institute, and the basic data are summarized in Table 1. Thirteen (54.1%) patients successfully weaned from PN and survived. Six (25.0%) patients kept HPN at the end of study period, and five (20.8%) patients expired without the discontinuation of PN. Fourteen (58.3%) of them were male, and the gender distribution revealed no difference among groups according to outcome. The age of onset was older in patients dependent on PN than those free from PN eventually; however, the statistical significance was unclear. Overall, the mean age of SBS or IF diagnosis was 355 days. The same trend was also observed in the BW, and the mean during disease onset was 5.15 kg. The BW percentile was smaller than 3 in most cases (83.3%). Prematurity was common (66.7%) in patients requiring HPN, and the percentage was higher in HPN dependence and mortality, although the *p* value was slightly over 0.05. Regarding low birth BW, no difference was observed among the groups.

### 3.2. Clinical Outcomes

The clinical outcomes and the PN mixture of HPN patients are listed in Table 2. A shorter follow-up period was observed in patients who failed to survive without statistical significance. The average duration of PN was 48.1 months. Unsurprisingly, the longest HPN period was observed in the HPN-dependent group. The longest period was 207 months in the current case series. Compared with HPN-free patients, patients who expired had 1.5 years longer life span with HPN. Among patients weaned HPN, the PN-free interval was 104.1 months on average. In the current case series, two patients had a temporary discontinuation of PN (1 and 5 months, respectively). However, weaning failed and the two patients still depended on HPN until the end of the study period. Significantly more episodes of hospitalization were noted in the mortality group, and the most was 55 times in 12 years. The longest interval of hospitalization was found in the HPN-dependent group, which was 7.12 months. The shortest was 2.57 months in the mortality group. The difference among the groups was significant (*p* = 0.021).

After HPN initiation, BW increased by an average of 25.6% and 25.4% per month within 3 and 6 months in the HPN-free group. The BW gain was slower after three months (17.7%) than within 3 months (24.7%) of HPN usage in the mortality group. Among the 24 patients, only one had no improvement by the BW percentile (>50 to <3). The monthly gain of BW was higher in the HPN-free and mortality groups than in the HPN-dependent group (9.7%).

The experience of commercial PN formula was limited for the pediatric patients. In the HPN-dependent group, the only patient had disease onset at the age of 14 years and was 22 years old by the end of study period. In the mortality group, a 12-year-old patient had the brief use of commercial PN for 3 months of the total HPN period (84 months).

### 3.3. SBS and IF

Table 3 lists the diagnoses of patients leading to SBS or IF and the related anatomy. The major indications for HPN included extended intestine resection (33.3%), congenital short bowel (33.3%), functional ileus with intestinal malabsorption (16.7%), multiple intestinal atresia (12.5%), and pancreatic juice leakage (4.1%). Among the above causes of short bowel, extended intestinal resection was significantly responsible for dependence on TPN (*p* = 0.014). Among patients who underwent extended resection (eight patients), extensive necrotizing enterocolitis was responsible for most cases (87.5%).

Regarding the anatomy of the residual intestine, the short residual length of the small intestine (<30 cm or <10% predicted length) and major colon resection (>50%) were correlated with HPN dependence and mortality (*p* = 0.003, both). Among five patients who failed to survive, four had short residual small-intestine length and major colon resection. The presence of an ileocecal valve was considered as beneficial to HPN-free survival, although the *p* value was exactly 0.05.

All patients had simultaneous enteral nutrition during the HPN period, except for one with gastroparesis requiring gastrostomy drainage. Only one patient in the current series received cadaveric intestine transplantation, Unfortunately, the patient developed rejection during hospitalization and expired eventually. Although other intestinal lengthening procedures, including STEP or the Bianchi operation [15] have been reported, no patients underwent such procedures in this series. For patients with congenital short bowel syndrome (33.3%) and intestinal malabsorption (16.7%), intestinal lengthening procedures are generally not demanded for weaning PN. For patients with extended intestinal resection, especially after the infancy period, the lengthening procedure was suggested. However, all the parents disagreed with the invasive surgery and preferred to wait for a transplant.

### 3.4. Central-Catheter-Related Complications

The results in Table 4 showed that every pediatric patient received 3.8 times central-line placement during the HPN period. In the HPN-dependent and mortality groups, most patients (90.9%) had three times of central-line placement or more. Overall, the mean lifespan of each central line catheter was 12.3 months. HPN dependence or mortality was unrelated to the subsequent risk of CRBSI, bacteremia, central-line dislodgement, occlusion, or rupture of the catheter. Meanwhile, none of the catheter-related complications contributed to mortality in the current series. The correlation between small bowel length and catheter-related infection times is plotted in Figure 1. One patient with 15 cm of residual small bowel had as high as 12 times of CVC infection, but the overall trend was unclear. The top three pathogens were *Staphylococcus*, *Klebsiella*, and *Candida species*.

### 3.5. Liver Function and Cholestasis

As shown in Table 5, cholestasis was associated with a higher rate of HPN dependence (*p* = 0.004) and mortality (*p* = 0.001). No difference existed between the average duration of cholestasis between the HPN-dependent group (1.4 months) and the HPN-free group (1.4 months). In the mortality group, the average duration of cholestasis of patients was significantly longer (19.3 months). The elevation of serum liver-function tests were common in all patients who required HPN. The duration of elevated serum AST or ALT was longer in the HPN-dependent group (46.3 months) than in HPN-free (4.2 months), and in the mortality (26.5 months) groups without statistical significance.

The Child–Turcotte–Pugh classification and Pediatric End-stage Liver Disease (PELD) score were used to evaluate the status of IFALD. One patient from the HPN-dependent group met the age criteria of the MELD score during evaluation and was excluded in the comparison. Patients in the mortality group were all Class B or C, and patients who survived were all classified as Class A (*p* < 0.0001). The PELD score was either zero or negative in the HPN-free group. Among the HPN-dependent patients, only one had a positive PELD score, and the mean was negative. All patients in the mortality group had a positive PELD score with a mean of 18.72. In the current series, a positive PELD score prohibited HPN-free survival (*p* < 0.0001).

### 3.6. Risk Factor Analysis

Results of the univariate and multivariate analyses of risk factors for HPN dependence and mortality are listed in Table 6 and Table 7, respectively. Univariate analysis demonstrated numerous significant factors contributing to the failure of PN weaning, such as extended intestinal resection, the absence of the ileocecal valve, more than two times of central-line insertion, cholestasis, Child Class B or C, and a positive PELD score. Three insignificantly contributing factors included small intestines shorter than 30 cm or <10% predicted length (OR = 16.20; *p* = 0.074), major colon loss (OR = 6.60; *p* = 0.053), and gastrointestinal tract bleeding or chronic anemia (OR = 5.83; *p* = 0.052). Multivariate logistic-regression analysis identified two independent risk factors for HPN dependence, extended resection (*p* = 0.032), and multiple central-line placement (*p* = 0.017). The regression model was validated according to the chi-squared test, and results confirmed the significance with a chi-squared of 10.773 and *p* = 0.013.

For the mortality group, the interval between admissions was shorter than 3 months. The anatomical features were unrelated to death, except for major colon loss (OR = 15.00; *p* = 0.030). Cholestasis, Child Class B or C, and positive PELD score were among the most significant risk factors of mortality. Patients presented with the complications of gastrointestinal tract bleeding, or chronic anemia requiring blood transfusion was another contributing factor of mortality. Other than Child Class B or C, the multivariate regression model identified the positive PELD score as the main risk factor of mortality. The regression model was validated via a chi-squared of 6.742 and *p* = 0.034.

### 3.7. Survival and Mortality

All five patients in the mortality group had hepatic failure. Three patients died from overwhelming sepsis, one from sudden cardiac arrest, and one from acute respiratory failure. The overall mortality rate in our series was 20.8% (5 out of 24 patients). Kaplan–Meier curves are presented in Figure 2. The 1, 5, 10, and 15-year overall survival rates were 95.8%, 71.5%, 73.6%, and 73.6%, respectively. The survival rates of patients dependent on HPN were 90.9%, 73%, 35.7%, and 35.7%, respectively. A 22-year probability of survival estimated according to the Kaplan–Meier curve was significantly lower than the calculated probability among children weaning from HPN (log-rank test, *p* = 0.003).

The Kaplan–Meier estimated survival rates of pediatric patients with HPN regarding different risk factors are illustrated in Figure 3. Both the lengths of colon and small intestine were related to survival according to the estimate. The tolerance criteria were 10% of the predicted length for the small intestine (log-rank test: *p* = 0.014) and 50% for the colon (log-rank test: *p* = 0.039), respectively. The common scoring systems for IFALD, such as the Child–Turcotte–Pugh Classification and the PELD score could successfully predict the survival of HPN patients. No patients with Child Class A (log-rank test: *p* < 0.0001) and a negative or zero PELD score (log-rank test: *p* < 0.0001) died after HPN usage.

## 4. Discussion

A comparison of patients receiving PN either at home or in the hospital reveals no significant differences in neonatal or clinical history between inpatients versus those at home, according to another cross-sectional study in Chile [16], which promotes long-term nutrition support in a home setting. In our study, more than half (54.1%) of children were successfully weaned from HPN, similar to another 14-year study with a result of 52% in France [17].

Patients with SBS are more likely to have Gram-negative infections, compared with other patients with CVC [18], whereas a shorter length of bowel is more likely to develop a catheter-related infection [19]. However, in our study, no significant correlation existed between the shorter length of the residual intestine and catheter-related infection (*p* = 0.263). The second and third most common pathogens of CRBSI were *Klebsiella* and *Candida* species. The etiology of such an infection is assumed to be related to the poor intestinal mucosal barrier, which is common in patients requiring PN.

A short residual bowel length was generally considered as an important prognostic factor for PN weaning. During the follow-up for the current case series, one 27-week preterm infant with necrotizing enterocolitis received a bowel resection at the age of 54 days. The initial residual length of the small intestine was 30 cm, and the patient was eventually weaned from HPN after 14 months. This case indicated that the predicted length percentage of resected or residual small bowel was important according to age. The age-predicted bowel length could be more accurate, considering a wide interindividual variation in anatomy [20]. The overall small intestinal length for infants between 19- and 27-week gestation is known to increase from 142 cm to 304 cm. For a comparable group of over 35-week gestation, the mean length of the small intestine is 275 cm at term, and 380 cm at 1 year [21,22]. In the present study, 10% of the residual small intestine (or 30 cm length for children older than 6 years) was set as the threshold for classification, considering the dynamic growth of the bowel through gestational development, rather than the absolute length. The importance of the colon length is infrequently addressed in the literature [15,16,17,20,23]. Herein, we found that the absence of an ileocecal valve (*p* = 0.022) had more impact on HPN dependence than major colon loss (*p* = 0.053). However, the major colon loss was related to mortality, more than ileocecal-valve deficiency (*p* = 0.030 and 0.256, respectively). The cutoff criterion defining colon loss was 50% of the total length, for the convenience of discussion.

No single risk factor is implicated as being causative of liver injury in PN-dependent patients [24]. A small bowel shorter than 100 cm has previously been reported as a significant independent variable for elevated serum liver-function tests (>1.5 of the upper limit of the normal range) probably owing to a higher parenteral caloric intake and an alteration in enterohepatic circulation [25]. The correlation between the length of the small bowel and the elevation of AST or ALT was not found in this study. Meanwhile, episodes of cholestasis, Child B or C, and positive PELD score were predictive variables for HPN dependence and mortality according to univariate risk-factor analysis.

Some studies have reported no difference in overall survival when patients are stratified according to PN status. However, the mortality rate was significantly higher in the HPN-dependent group (45.4%, *p* = 0.006) in our study. All mortality cases suffered from hepatic failure, and the major death cause was sepsis in the current work. Similarly, the literature has also identified deaths from liver disease and sepsis, with HPN-related mortality rates ranging within 38–48% [23,26].

Central-line-associated bloodstream infections (CLABSIs) are traditionally considered as a significant cause of morbidity and mortality in those with IF, and depending on PN [27,28]. The correlation between central-line infection and HPN dependence or mortality was not found in the current study. Other complications such IFALD, venous thrombosis, and delayed intestinal adaptation contributed to by CLABSIs [27,29,30], may be evaluated through qualitative or quantitative analysis with an image study integrated in the future.

This study had some limitations. First, it was performed in a single medical center, thereby limiting the generalizability of the results. Second, the sample size of our retrospective study was small and may enhance the bias in the statistical results. However, the long follow-up period for observation in this study could improve the validation of the contributory findings.

## 5. Conclusions

More than half of the pediatric patients were successfully weaned off HPN after IF within the past 22-year follow-up period. Caution is warranted to prevent cholestasis and IFALD during the HPN period, to prevent mortality. Extended bowel resection and multiple central-line insertion were independently related to HPN dependence. By understanding the risks of HPN dependence and mortality, preventive procedures could be addressed earlier.

## Figures and Tables

**Figure 1 nutrients-15-00706-f001:**
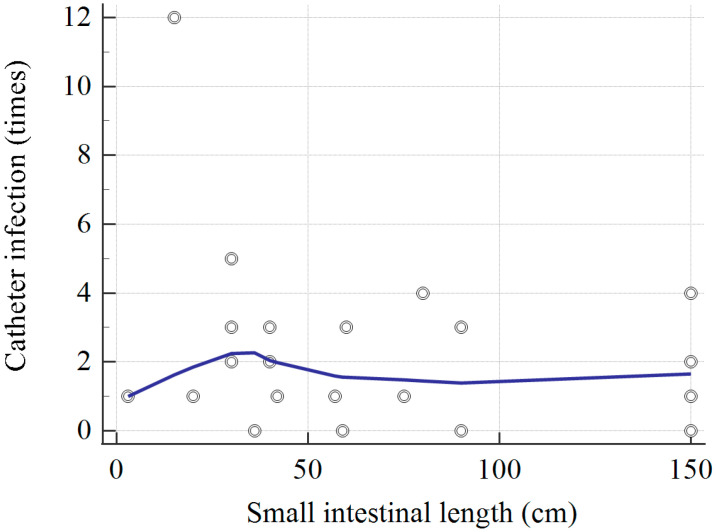
The scatter diagram between small intestinal length and catheter-related infection times. Correlation coefficient, r = −0.237 (−0.584 to 0.183) and *p* = 0.263. Trend line by locally weighted scatterplot smoothing and span = 60%.

**Figure 2 nutrients-15-00706-f002:**
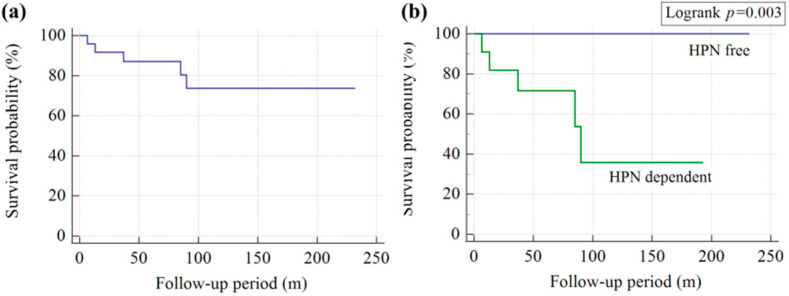
(**a**) Kaplan–Meier survival curve of all-cause mortality in HPN patients. (**b**) Comparison between HPN-free and HPN-dependent patients (HPN-dependent and mortality groups). It was noted than the overall survival was significantly longer in HPN-free patients than in HPN-dependent patients (logrank test: *p* = 0.003).

**Figure 3 nutrients-15-00706-f003:**
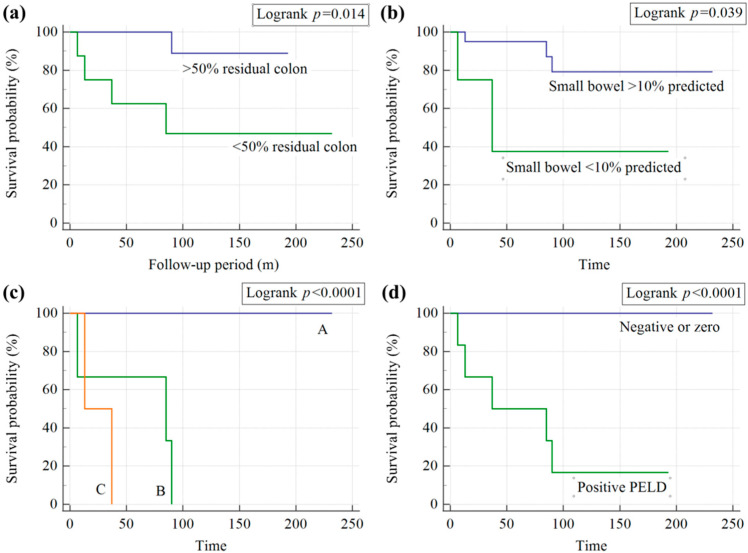
Kaplan–Meier estimated survival rates of pediatric patients with HPN, considering the different risk factors: (**a**) major colon loss (>50%), (**b**) residual small intestine length <10% predicted, (**c**) Child-Turcotte-Pugh Classification, and (**d**) Pediatric End-stage Liver Disease (PELD) score. A, B, and C stand for Child Class A, B, and C, respectively.

**Table 1 nutrients-15-00706-t001:** Demographic characteristics of pediatric HPN patients.

	HPN Free	HPN Dependent	Mortality	*p*
Patient number	13	6	5	
Female	4 (30.7%)	3 (50.0%)	3 (60.0%)	0.472
Age of disease onset (days, mean)	1–90 (34.2)	6–5340 (905.0)	1–2520 (529.8)	0.316
Prematurity	6 (46.1%)	6 (100.0%)	4 (80.0%)	0.053
Low BBW (<2500 g)	8 (61.5%)	4 (66.7%)	3 (60.0%)	0.969
BW of disease onset (kg, mean)	0.8–4.3 (2.64)	2.5–4.5 (9.94)	2.0–19.0 (5.95)	0.275
BW percentile				
50–75 25–50 10–25 3–10 <3	1 0 1 0 11	0 0 0 1 5	0 0 1 0 4	0.643 - 0.485 0.209 0.972

Abbreviations: BBW, birth body weight; BW, body weight.

**Table 2 nutrients-15-00706-t002:** Clinical outcomes and formula of pediatric HPN patients.

	HPN Free (13)	HPN Dependent (6)	Mortality (5)	*p*
HPN period (month, mean)	5–115 (29.0)	19–207 (91.3)	7–89 (46.0)	0.045
Hospitalization times (mean)	1–24 (6.5)	4–39 (14.2)	4–55 (20.2)	0.006
Interval of hospitalization (month, mean)	2.1–9.0 (4.60)	3.2–12.8 (7.12)	1.5–4.1 (2.57)	0.021
Follow-up period (month, mean)	26–232 (115.8)	24–193 (99.3)	7–90 (46.3)	0.097
Commercial admixture usage	0	1 (16.6%) *	1 (20%) **	0.270
BW gain by month in 3 months after HPN (%, mean)	25.6%	19.6%	24.7%	0.715
BW gain by month in 6 months after HPN (%, mean)	25.4%	17.3%	17.7%	0.448
BW gain after HPN (%, mean)	347.9%	493.5%	356.1%	0.666
BW gain after HPN by month (%, mean)	15.2%	9.7%	14.1%	0.691

* Abbreviations: The patient had disease onset at the age of 14 years. ** The 12-year-old patient had brief use for 3 months of HPN period (84 months).

**Table 3 nutrients-15-00706-t003:** Major causes and related anatomies of SBS or IF of pediatric HPN patients.

	HPN Free (13)	HPN Dependent (6)	Mortality (5)	*p*
Congenital short bowel	5	2	1	0.758
Multiple intestinal atresia	2	0	1	0.545
Extended intestinal resection	1	4	3	0.014
Intestinal malabsorption	4	0	0	0.131
Pancreatic leak	1	0	0	0.643
Small intestines length (cm, mean)	30–150 (75.7)	3–150 (58.0)	20–75 (35.0)	0.236
<30 cm (<10% predicted)	2	2	4	0.033
30–90 cm (10–30%)	8	3	1	0.287
>90 cm (>30%)	3	1	0	0.500
Absence of IC valve	2	4	3	0.050
Majority of colon (>50%) loss	2	2	4	0.033

Abbreviations: IC, ileocecal valve.

**Table 4 nutrients-15-00706-t004:** Central line catheter-related complications of HPN patients.

	HPN Free (13)	HPN Dependent (6)	Mortality (5)	*p*
Central line placement (mean)	1–4 (2.3) 11.3 catheter-month	3–23 (7.0) 15.7 catheter-month	1–9 (4.0) 10.9 catheter-month	0.098
1 time	4	0	1	0.307
2 times	4	0	0	0.131
3 times	2	3	2	0.254
>3 times	3	3	2	0.636
Infection (mean)	0–4 (1.5) 19.8 catheter-month	0–12 (3.8) 23.8 catheter-month	1–5 (2.6) 17.7 catheter-month	0.175
Occlusion (mean)	0–1 (0.1)	0–3 (0.8)	0–1 (0.2)	0.116
Dislodgement (mean)	0–1 (0.2)	0–3 (0.7)	0–5 (1.0)	0.363
Rupture (mean)	0–1 (0.5)	0–7 (1.7)	0–1 (0.2)	0.056

**Table 5 nutrients-15-00706-t005:** Liver function and cholestasis of HPN patients.

	HPN Free (13)	HPN Dependent (6)	Mortality (5)	*p*
Cholestasis *	2	3	5	0.004
Duration (m)	0–17 (1.4)	0–4 (1.4)	2–41 (19.3)	0.001
Elevated AST/ALT **	8	4	5	0.370
Duration (m)	0–36 (4.2)	0–166 (46.3)	6–41 (26.5)	0.057
Child-Turcotte-Pugh classification				<0.0001
A	13	6	0	
B	0	0	3	
C	0	0	2	
PELD score *** (mean)	−16.6–0.0 (−10.18)	−5.0–15.0 (−1.60)	8.4–24.6 (18.72)	<0.0001
Positive PELD score ***	1	5	5	<0.0001

Abbreviations: AST, aspartate aminotransferase; ALT, alanine transaminase; PELD, Pediatric End-Stage Liver Disease. * Defined as any time conjugated bilirubin ≥2.5 mg/dL during HPN period. ** Defined as any value ≥ two times the upper limit during HPN period. *** One patient in HPN dependent group that was not evaluated by PELD score was excluded.

**Table 6 nutrients-15-00706-t006:** Risk factor analysis for HPN dependence.

	**Univariate Analysis**		**Multivariate Analysis**		
**Variables**	**OR (95% CI)**	** *p* **	**Coefficient**	**SE**	** *p* **
Male	0.37 (0.06–1.97)	0.244			
BW < 3 percentile	0.48 (0.06–3.61)	0.479			
Neonatal onset	3.11 (0.55–17.33)	0.195			
Prematurity	1.22 (0.14–10.48)	0.854			
Short admission internal (<3 m)	2.06 (0.27–5.35)	0.479			
Extended resection	14.40 (1.35–152.53)	0.026	0.4760	0.2044	0.032
Small intestine < 10% predicted	16.20 (0.76–343.81)	0.074			
No IC valve	9.62 (1.37–67.24)	0.022	−0.1420	0.2283	0.542
Colon loss (>50%)	6.60 (0.96–44.92)	0.053			
Central line > 2 times	16.00 (1.54–166.05)	0.005	0.4298	0.1633	0.017
Cholestasis	32.00 (2.80–364.79)	0.020	0.1066	0.2749	0.703
Child B or C	22.84 (1.09–478.83)	0.043	0.3148	0.3805	0.419
Positive PELD	31.90 (1.52–668.78)	0.025	0.1747	0.3881	0.658
GI bleeding or chronic anemia	5.83 (0.98–34.64)	0.052			

Abbreviation: OR, odds ratio; CI, confidence interval; SE, standard error; BW, body weight; IC, ileocecal; PELD, Pediatric End-stage Liver Disease score; GI, gastrointestinal.

**Table 7 nutrients-15-00706-t007:** Risk factor analysis for mortality.

	Univariate Analysis		Multivariate Analysis		
Variables	OR (95% CI)	*p*	Coefficient	SE	*p*
Male	0.38 (0.05–2.92)	0.358			
BW < 3 percentile	1.06 (0.09–12.40)	0.958			
Neonatal onset	1.09 (0.14–8.12)	0.932			
Prematurity	1.33 (0.10–16.48)	0.822			
Short admission internal (<3 m)	12.75 (1.26–128.78)	0.031	0.0873	0.1269	0.5002
Extended resection	1.86 (0.23–14.64)	0.552			
Small intestine < 10% predicted	5.66 (0.56–57.23)	0.141			
No IC valve	3.25 (0.42–24.84)	0.256			
Colon loss (>50%)	15.00 (1.29–174.39)	0.030	0.1278	0.1128	0.2720
Central line > 2 times	2.90 (0.27–31.21)	0.377			
Cholestasis	37.88 (1.74–823.97)	0.020	−0.0009	0.1278	0.9939
Child B or C *	429.00 (7.60–24198.81)	0.003	1.0000 **	0	-
Positive PELD	135.66 (4.81–3825.41)	0.004	0.7691	0.1735	0.0003
GI bleeding or chronic anemia	29.00 (1.36–616.62)	0.030	−0.0330	0.1233	0.7917

Abbreviation: OR, odds ratio; CI, confidence interval; SE, standard error; BW, body weight; IC, ileocecal; PELD, Pediatric End-stage Liver Disease score; GI, gastrointestinal. * Child B or C was not included in the multivariate regression because of the overfitting of model. ** All patients of mortality group had Child Class B or C, and all survived patients were Child A.

## Data Availability

The data are unavailable due to the policy of the institution’s IRB.

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
