# Peer review of "Home Parenteral Nutrition for Children: What Are the Factors Indicating Dependence and Mortality?"

_nutrients, 2023, doi:10.3390/nu15030706_

Round 1
Reviewer 1 Report
Dear Authors
This is a very informative paper on this topic. I would review the references in order to update them if you think it is appropriate
Author Response
Thank you for your opinions. We checked the cited references and hoped to get your comments in detail.
Reviewer 2 Report
This is the analysis of 24 cases of pediatric intestinal failure.
The number of participants is very limited, there is a number of larger series presented so far.
I have some major comments:
1. The HPN-dependency is associated with the quality of intestinal rehabilitation such as early enetral feeding introduction, we have no data what procedure you use in your center,
2. The parenteral admixture composition, and the lipid emulsion is also crucial for the weaning and intestinal rehabilitation. We have no data on parenteral admixture composition and the types of formulation used.
3. Infwctious complicatins: what is the definition of CRBSI, whal was the ethiology of infections?
Author Response
Thank you for your valuable opinions. The authors acknowledged that the series is small and listed the limitations in the final portion of the discussion (lines 362-366). Please see the attachment for responses to the comments.

Round 2
Reviewer 2 Report
The authors answered my comments.